# Hierarchical Attention: What Really Counts in Various NLP Tasks

## Abstract

Attention mechanisms in sequence to sequence models have shown great ability and wonderful performance in various natural language processing (NLP) tasks, such as sentence embedding, text generation, machine translation, machine reading comprehension, etc. Unfortunately, existing attention mechanisms only learn either high-level or low-level features. In this paper, we think that the lack of hierarchical mechanisms is a bottleneck in improving the performance of the attention mechanisms, and propose a novel **Hierarchical Attention Mechanism (Ham)** based on the weighted sum of different layers of a multi-level attention. **Ham** achieves a state-of-the-art BLEU score of $0.26$ on Chinese poem generation task and a nearly $6.5\%$ averaged improvement compared with the existing machine reading comprehension models such as BIDAF and Match-LSTM. Furthermore, our experiments and theorems reveal that **Ham** has greater generalization and representation ability than existing attention mechanisms.

## 1 Introduction

In recent years, long short-term memory, convolutional and recurrent networks with attention mechanisms have been very successful on a variety of NLP tasks. Most of the tasks In NLP can be formulated as sequence to sequence problems. Moreover, encoders and decoders are vital components in sequence to sequence models while processing the inputs and outputs.

An Attention mechanism works as a connector between the encoders and decoders and help the decoder to decide which parts of the source text to pay attention to. Thus an attention mechanism can integrates sequence models and transduction models, so it is able to connect two words in a single passage or paragraph without regard to their positions.

Attention mechanisms have become integral components in various NLP models. For example, in machine translation tasks, attention mechanism-based models [1, 2, 3] have ever been the state-of-the-art; in sentence embedding, self attention based model is now the state-of-the-art [4]; in machine reading comprehension, almost every recently-published model, such as BIDAF[5], Match-LSTM[6], Reinforcement Ranker Reader[7], and R-NET[8], contains attention mechanism; in abstractive summarization model [12] which has also once been the state-of-the-art, attention mechanism is very necessary, and in poem generation [9], attention mechanism is also widely used. More surprisingly, Vaswani, et al. (2017) showed that their model **Transformer** which relies solely on the attention mechanisms can outperform those RNN or LSTM-based existing models in machine translation tasks. Thus, they stated that "Attention is all you need".

However, we note that the potential issue with the existing attention mechanisms is that the basic attention mechanism learns only the low-level features while the multi-level attention mechanism learns only the high-level features. This may make it difficult for the model to capture the intermediate feature information, especially when the source texts are long. In order to address this issue, we present **Ham** which introduces a hierarchical mechanism into the existing multi-level attention mechanisms. Each time when we perform a multi-level attention, instead of using the result of the last attention level only, we use the weighted sum of the results of all the attention levels as the final output.

We show that **Ham** can learn all levels of features among the tokens in the input sequence and give a proof of its monotonicity and convergency. This work presents the design and implementation of

**Ham** and our implementation performs well on a range of tasks by replacing the existing attention mechanisms in different models of different tasks.

We are able to achieve results comparable to or better than existing state-of-the-art model. On Chinese poem generation, our model scores $0.246$ BLUE, an improvement of $21.78\%$ from a RNN-based Poem Generator model. On machine reading comprehension task, our implementation is more effective, model with **Ham** has achieved an average improvement of $6.5\%$ compared to previous models.

The implementation of the Hierarchical Attention Mechanism is not difficult and the code will be available on http://github.com after the acceptance.

## 2 ATTENTION MECHANISMS

The attention mechanism can be described as a function whose input is a query and a set of key-value pairs, where the query and keys are vectors with the same dimension (denoted $d_k$), and the values are defined as $d_v$-vectors. Note that in most types of attention mechanisms, the values are equal to the keys. Through the mapping of the attention mechanism, the input can be mapped to a single vector, which is as the output.

### 2.1 THE VANILLA ATTENTION MECHANISM(VAM)

Given a query $\mathbf{q} \in \mathbb{R}^{d_k}$ and an input sequence $K = [\mathbf{k}_1, \mathbf{k}_2, \ldots, \mathbf{k}_n] \in \mathbb{R}^{d_k \times n}$ where $\mathbf{k}_i \in \mathbb{R}^{d_k}$ denotes the word embedding of the $i$-th word of the sequence, the vanilla attention mechanism aims at using a compatibility function $f(\mathbf{k}_i, \mathbf{q})$ to compute a relativity score between the query $\mathbf{q}$ and each word $\mathbf{k}_i$. This score is treated as the attention value of $\mathbf{q}$ to $\mathbf{k}_i$. Then we have $n$ attention scores $f(\mathbf{k}_i, \mathbf{q})$ for $i = 1, 2, \ldots, n$. Now we apply the softmax function to define a categorical distribution:

$$p(i|K, \mathbf{q}) = \mathrm{softmax}(f(\mathbf{k}_i, \mathbf{q})) = \frac{\exp(f(\mathbf{k}_i, \mathbf{q}))}{\sum_{j=1}^{n} \exp(f(\mathbf{k}_j, \mathbf{q}))}.$$

Futher, we compute the output which is represented as the weighted sum of the input sequence:

$$\mathbf{s} = \sum_{i=1}^{n} p(i|K, \mathbf{q})\mathbf{k}_i.$$

The attention mechanism above is the original version which was firstly proposed by Bahdanau, et al. (2014). In the **Scaled Dot-product Attention Mechanism**, the compatibility function $f$ is defined as the scaled dot product function $f(\mathbf{k}_i, \mathbf{q}) = \frac{<\mathbf{k}_i, \mathbf{q}>}{\sqrt{d_k}}$. Here the scaling factor $\frac{1}{\sqrt{d_k}}$ is used to prevent the dot product from growing too large in magnitude.

### 2.2 SOFT, HARD AND LOCAL ATTENTION MECHANISMS

There are three different types of mechanisms: soft, hard and local. The main difference between them is the region where attention function is calculated. The **VAM** belongs to soft attention where the categorical distribution $p(i|K, \mathbf{q})$ is computed over the whole input sequence of words. Thus it is also referred to as global attention. The resulting distribution can reflect the relatedness or importance between the query and every word in the input sequence, and we use these importance scores as the weights of the output sum. Soft attention takes every word in the input sequence, no matter what kind of word it is, into consideration. Soft attention is differentiable and parameter-free, but is computationally expensive and less accurate.

In order to overcome the weakness of soft attention, hard attention is a natural alternative. Contrary to the widely-studied soft attention, hard attention locates accurately to only one key $k_{i_0}$. In other words, the probability of getting the special key $k_{i_0}$ is 1 and others be 0. This implies that the choice of the one key means everything to the performance of the model. The action of choosing is not differentiable, so one uses reinforcement learning methods instead, such as policy gradient method.

As we have seen, soft and hard attentions are two extreme cases. Xu, et al. (2015) proposed a hybrid attention mechanism. Instead of choosing every key or only one key, one chooses a subset of all the

keys from the input sequence. When computing the attention, one can just focus on the important part of the keys and discard the rest, thus it is also referred to as local attention. This attention mechanism combines the wideness and accuracy when choosing keys. The subset-choosing process is non-differentiable and reinforcement learning methods are also needed.

## 2.3 MULTI-HEAD ATTENTION AND MULTI-LEVEL ATTENTION MECHANISMS

The multi-head attention mechanism proposed by Vaswani, et al. (2017) plays an important role in the **Transformer** model which is state-of-the-art in neural machine translation. Instead of calculating a single attention function with queries, keys and values, it linearly projects the queries, keys and values $h$ times to $d_k$, $d_k$ and $d_v$ dimensions, respectively. On each version of linear projections, the attention function is performed in parallel and yields several versions of $d_v$-dimensional scaled dot-product attentions. Subsequently, these attention values are concatenated and once again projected, resulting in the final value as the output of the multi-head attention mechanism. That is,

$$\text{Attention}(Q, K, V) = \text{softmax}(\frac{QK^T}{\sqrt{d_k}})V,$$

$$\text{MultiHead}(Q, K, V) = \textsf{Concat}(\text{head}_1, \cdots, \text{head}_h)W^O,$$

where $\text{head}_i = \text{Attention}(QW_i^Q, KW_i^K, VW_i^V)$ and $W_i^Q, W_i^K, W_i^V, W_i^O$ are all projection matrices.

The multi-level attention mechanism is another variety of attention mechanisms. Instead of increasing the number of heads, the multi-level attention increases the number of levels. For example, in a two-level attention mechanism, we calculate the attention value of the query $\mathbf{q}$ and the keys, which is represented as $\text{Attention}(\mathbf{q}, K, K)$. This output has the same dimension as the query, giving the first level. In the second level, we treat the output as the new query and calculate the attention value with the input sequence (or keys) $K$ again. The result can be represented as $\text{Attention}(\text{Attention}(\mathbf{q}, K, K), K, K)$. The second attention can learn a higher level of internal features among the words of the input text.

Based on the long line of previous attempt, Cui et al. (2017) proposed a novel way of treating various documents in neural machine translation. They used a self-attention mechanism to encode the words in every document and then used a second attention over different documents to learn a higher level of features among the words of different documents. Yang, et al. (2016) also proposed a 2-level hierarchical attention network for document classification task. In recently proposed **Transformer** mode [1], the authors repeated the attention mechanism $N$ times over the input sequence in order to learn the higher level feature. This is an $N$-level attention mechanism through which the input sequences can be changed again and again into sequences much more suitable for feature extraction and decoder input. This is why so-called **Transformer**.

## 2.4 SELF ATTENTION MECHANISM

In the self attention mechanism the query and key are the same. In other words, the query $\mathbf{q}$ stems from the input sequence $K$ itself. Using self attention mechanism, we are able to learn the relatedness of different parts of the input sequence, no matter what their distance is. With self-attention, a long text can be encoded to a more suitable input for the decoder. Similar to the original attention mechanism, self-attentions also have three different types: soft, hard and local, as well as have multi-head version and multi-level version.

## 3 HIERARCHICAL ATTENTION MECHANISM (HAM)

In this section we present two Hierarchical Attention models built on the vanilla attention and self attention, respectively.

### 3.1 HIERARCHICAL VANILLA ATTENTION MECHANISM (HAM-V)

We have mentioned above that multi-level attention mechanisms can learn a deeper level of features among all the tokens of the input sequence and the query. In our model, we use multi-level for

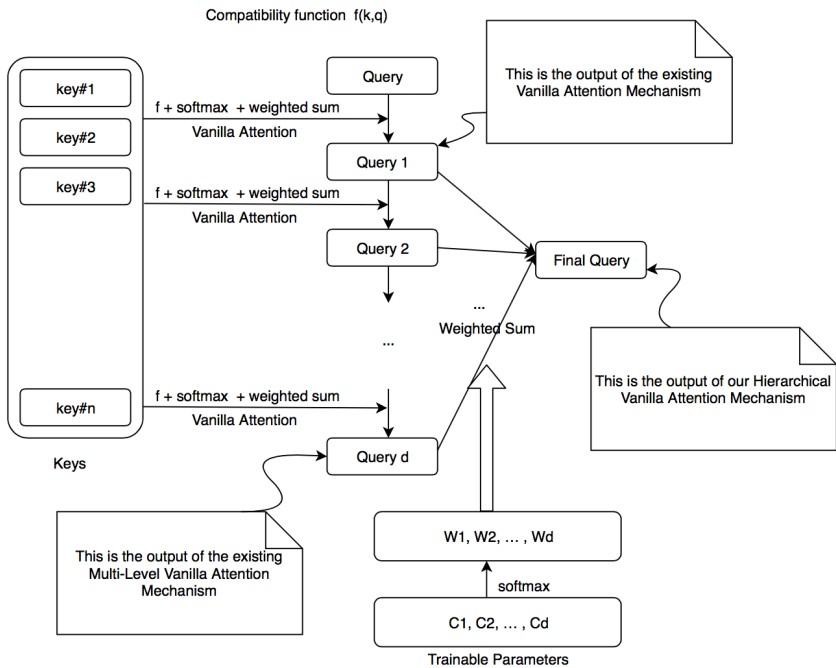

Figure 1: Hierarchical Vanilla Attention Mechanism

reference but different from the existing **Multi-Level Attention Mechanisms**. Our **Ham-V** focus on all the intermediate attention results rather than just the result of the last attention level. As shown in Figure1, given the query and the input sequence which consists of $n$ keys, we calculate the **Vanilla Attention Mechanism** result of them and get Query 1. And then we continue to calculate the attention result of Query 1 and the keys and get Query 2. Repeat this calculation $d$ times. Thus, we form a $d$-depth attention. Finally, the output of our **Ham-V** is the weighted sum of the above $d$ attention results, where the $d$ weights are the softmax values of $d$ trainable parameters. The softmax is used to convert these $d$ weights into the probabilities. These weights can tell us the relative importance of the $d$ intermediate attention results Query $i$. In other words, the relative importance of the $d$ attention levels.

### 3.2 HIERARCHICAL SELF ATTENTION MECHANISM (HAM-S)

In **Self Attention Mechanisms**, the query stems from the input sequence **k** itself. So it can be treated as a special case of attention mechanisms. Similarly, the self-version of **Hierarchical Attention Mechanisms**, which is shown in Figure 2, takes only the sequence **k** as input. We calculate the self-attention results of the input sequence for $d$ times consecutively. Finally, the output of our **Ham-S** is the weighted sum of these $d$ attention results, where the $d$ weights are the softmax values of $d$ trainable parameters $w_1, w_2, \cdots, w_d$ which is the same as **Ham-V**. Through the $d$ levels of self-attention, our model can learn different levels of deep features among all the tokens of the input sequence, and through the $d$ trainable parameters and the weighted sum mechanism, our model can learn the relative importance of the $d$ self-attention levels.

## 4 ANALYSIS

We present theoretical analysis of our **Ham** mainly in two aspects: representation ability and convergence. The representation ability of **Ham** is obviously higher than **Vanilla Attention Mechanisms** and **Multi-Level Attention Mechanisms**. Because the two attention mechanisms are just two extreme cases of **Ham**. When we set $w_1 = 1$, $w_2 = w_3 = \cdots = w_d = 0$, our **Ham** model is equivalent to the former. When we set $w_d = 1$, $w_1 = w_2 = \cdots = w_{d-1} = 0$, our model is equivalent to the latter. Thus **Ham** is much more general and the representation ability is much higher. Using the

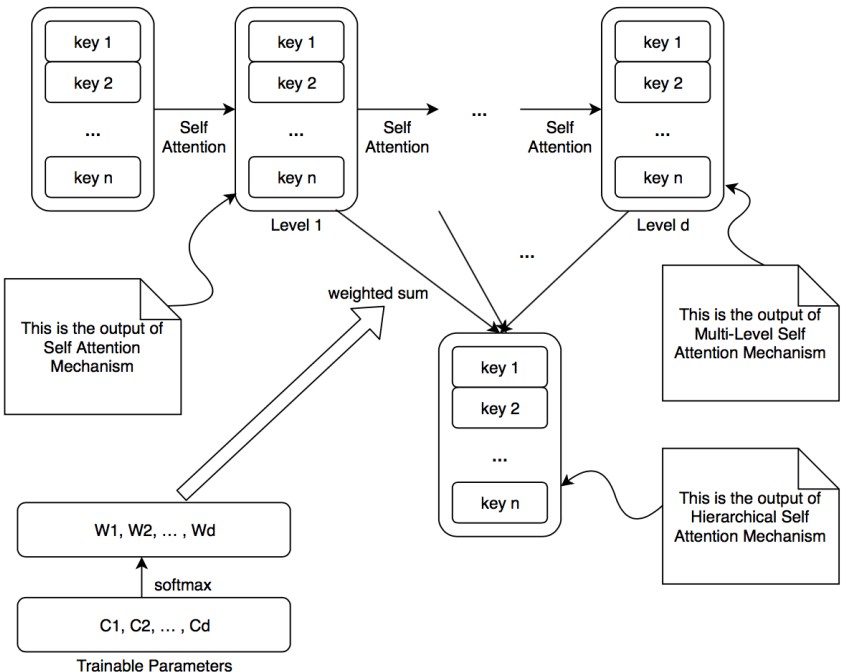

Figure 2: Hierarchical Self Attention Mechanism

weighted linear combination of these $d$ intermediate attention results, our model takes every level of features into consideration.

We are going to prove that the global minimum value of our loss function $L(\mathrm{Ham}(c_1, \cdots, c_d), X, \theta)$ of the whole **Ham** model will decrease monotonically and converges finally as the increase of hierarchical attention depth $d$. Here, $\mathrm{Ham}(c_1, \cdots, c_d)$ denotes a **Ham** with the attention depth $d$ and $d$ trainable parameters $c_1, \cdots, c_d$, $X$ denotes all the input data including the queries and keys, and $\theta$ denotes all the parameters in the other part of the whole model. It is also worth emphasizing that all loss functions in NLP tasks have positive values.

**Theorem 1.** *In Vanilla Attention Mechanism, we have that*

$$\min_{1 \leqslant i \leqslant n} \|\mathbf{k}_i\|_2 \leqslant \|\mathrm{Attention}(\mathbf{q}, K, K)\|_2 \leqslant \max_{1 \leqslant i \leqslant n} \|\mathbf{k}_i\|_2.$$

This result is very obvious according to the definition of **Vanilla Attention Mechanism**, $\mathrm{Attention}(\mathbf{q}, K, K)$ is the weighted sum of $\mathbf{k}_1, \mathbf{k}_2, \cdots, \mathbf{k}_n$ and the weights are nonnegative with their sum 1. Denote the weights as $\alpha_1, \cdots, \alpha_n$. Then

$$\|\mathrm{Attention}(\mathbf{q}, K, K)\|_2 = \Big\| \sum_{i=1}^{n} \alpha_i \mathbf{k}_i \Big\|_2 \leqslant \sum_{i=1}^{n} \alpha_i \|\mathbf{k}_i\|_2 \leqslant \max_{1 \leqslant i \leqslant n} \|\mathbf{k}_i\|_2.$$

The left-hand side of the inequality can be proved similarly. This theorem tells us that through multiple attention layers, the vectors of intermediate attention levels will neither explode nor vanish as the increase of attention depth $d$.

**Theorem 2.** *Let* $A_d = \min\limits_{c_i, \theta} L(\mathrm{Ham}(c_1, \cdots, c_d), X, \theta)$ *be the global minimal value of the loss function. Then* $\{A_d\}|_{d=1}^{+\infty}$ *is a monotonically decreasing sequence and it will converge.*

It is easy to note that:

$$
\begin{aligned}
A_d &= \min_{c_i(1 \leqslant i \leqslant d), \theta} L(\mathrm{Ham}(c_1, \cdots, c_d), X, \theta) \\
&= \min_{c_i(1 \leqslant i \leqslant d), \theta} L(\mathrm{Ham}(c_1, \cdots, c_d, c_{d+1} = -\infty), X, \theta) \\
&\geqslant \min_{c_i(1 \leqslant i \leqslant d+1), \theta} L(\mathrm{Ham}(c_1, \cdots, c_d, c_{d+1}), X, \theta) = A_{d+1}.
\end{aligned}
$$

This means the monotonicity of the sequence $\{A_d\}|_{d=1}^{+\infty}$. On the other hand, since our loss function always has positive values, sequence $\{A_d\}$ has 0 as its lower bound. Therefore, the monotonically decreasing sequence $\{A_d\}|_{d=1}^{+\infty}$ will converge.

## 5    EXPERIMENTS

Attention mechanisms are widely used in various NLP tasks. In our experiments, we would like to replace existing attention mechanisms with our novel **Ham-V** and replace existing self-attention mechanisms with our **Ham-S** to show the powerfulness and generalization ability of our model. We conduct our experiments on two different NLP tasks, Chinese Poem Generation and Machine Reading Comprehension.

### 5.1    MACHINE READING COMPREHENSION

The first NLP task we used to test our **Ham** is the machine reading comprehension (MRC). We conduct our experiment on both English MRC and Chinese MRC. The baseline models we use include BIDAF[5], Match-LSTM[6], R-NET[8]. Here, BIDAF is for both Chinese and the other two are for English. They have a major similarity that all of them use attention mechanism as the connection between their encoders and decoders and they are all open source. Their code is available on `http://github.com/baidu/DuReader`, `https://github.com/MurtyShikhar/Question-Answering` and `https://github.com/NLPLearn/R-net`. What we will do is to replace their attention mechanism with **Ham** and compare the difference of their performance. Specially, the R-NET model contains two attention mechanisms when doing question-passage matching and passage self-matching. Here, we replace them both with **Ham** and **Ham-S**.

The Chinese dataset we use for MRC experiments is DuREADER which is introduced by He, et al. (2017) and the English dataset we use include SQUAD which can be downloaded from `https://rajpurkar.github.io/SQuAD-explorer` and MS-MARCO which can be downloaded from `http://www.msmarco.org`. We randomly choose 10 percent of question-answer data as testing set and the rest as training set. The evaluation method we use is BLEU-4 and ROUGE-L for Chinese, ExactMatch(EM) and F1 score for English, where EM measures the percentage of how much the prediction of the model matches ground truth exactly and F1 measures the overlap between prediction and ground truth. During our experiments, we set also different attention depths $d$ to show the influence of attention depth.

### 5.2    CHINESE POEM GENERATION

In this work we generate Chinese quatrains, each of whose lines has the same length of 5 or 7 characters. The baseline model we use is **Planning based Poetry Generation (PPG)** proposed by Wang, et al. (2016), which generates Chinese poetries with a planning based neural network. Once we input a Chinese text, this model will generate a highly-related Chinese quatrains as follows.

Firstly, the model extracts keywords from this input text with **TextRank algorithm** proposed by Mihalcea, et al. (2004). Next, if the number of extracted keywords is not enough for a whole quatrain, more keywords will be created by **Knowledge-based method**. Then it comes to the final step, poem generation. The quatrain is generated line by line and each line corresponds a keyword. When generating a single line, one uses a bi-directional Gated Recurrent Unit (GRU) model proposed by Cho, et al. (2014 ) as encoder and another GRU model as decoder. Between encoder and decoder, an attention mechanism is used for connection.

It is worth emphasizing that the dataset used by **PPG** consists of 76,859 quatrains from the Internet and **PPG** randomly chooses 2000 quatrains for testing, 2000 for validation and the rest for training. In the encoder part of **PPG**, the word embedding dimensionality is set as 512 and initialized by word2vec (Mikolov, et al. (2013)). In both GRU models, the hidden layers also contain 512 hidden units but they are initialized randomly. For more details, please read Wang, et al. (2016). The code and dataset of **PPG** model can be found from `https://github.com/Disiok/poetry-seq2seq`.

| Model | Dataset | BLEU-4 | ROUGE-L |
|---|---|---|---|
| BIDAF | DuReader | 33.95 | 44.20 |
| BIDAF with 2-level **Ham** | DuReader | 34.02 | 44.39 |
| BIDAF with 5-level **Ham** | DuReader | 34.79 | 45.10 |
| BIDAF with 10-level **Ham** | DuReader | **35.96** | 47.33 |
| BIDAF with 20-level **Ham** | DuReader | 35.79 | **47.41** |
| Model | Dataset | EM | F1 |
| Match-LSTM | SQUAD | 54.29 | 66.87 |
| Match-LSTM with 2-level **Ham** | SQUAD | 54.41 | 66.99 |
| Match-LSTM with 5-level **Ham** | SQUAD | 55.29 | 69.03 |
| Match-LSTM with 10-level **Ham** | SQUAD | 58.37 | 70.70 |
| Match-LSTM with 20-level **Ham** | SQUAD | **58.47** | **70.81** |
| Model | Dataset | BLEU-1 | ROUGE-L |
| R-NET | MSMARCO | 41.29 | 43.38 |
| R-NET with 2-level **Ham** | MSMARCO | 41.37 | 43.89 |
| R-NET with 5-level **Ham** | MSMARCO | 43.13 | 45.67 |
| R-NET with 10-level **Ham** | MSMARCO | 43.48 | 45.75 |
| R-NET with 20-level **Ham** | MSMARCO | **43.62** | **45.78** |

Table 1: Evaluation results for MRC models

In our experiment, we replace the attention part of **PPG** from a Vanilla Attention Mechanism to our **Ham-V** and set the compatibility function $f$ to be the scaled dot product function, while keeping other parts and dataset unchanged as **PPG** except the evaluation part. The evaluation of poem generation in **PPG** is done by experts and we can not keep evaluation method unchanged since it is not convincing to find experts for evaluation. Our evaluation algorithm is based on BLEU-2 score which is calculated as

$$\text{BLEU} = \frac{1}{3} \sum_{i=1}^{3} \text{BLEU}_i,$$

where $\text{BLEU}_i$ denotes the BLEU-2 score computed for the next $(i + 1)$th lines given the previous $i$ goldstandard lines. This averaged BLEU can tell us how much correlated the lines of a generated quatrain are. We will show some quatrains generated by our **Ham-based PPG** in the appendix.

## 6 EXPERIMENTAL RESULTS AND QUALITATIVE ANALYSIS

### 6.1 MACHINE READING COMPREHENSION

As clearly visible in Table 1, the proposed model is much better than conventional model at Chinese and English machine reading comprehension. This is likely due to the fact that our human language has a kind of hierarchical relationship within itself both structurally and semantically. And our hierarchical attention mechanism is much easier to capture the inherent structural and semantical hierarchical relationship in the source texts because of their innate similarity. We also set the attention depths $d$ to be 1 (which is equivalent to ordinary models without using **Ham**), 2, 5, 10 and 20.

From Table 1, we can find that our **Ham** plays a significant role of the whole model. With the increase of attention depth $d$, the performance rises quickly at first and starts to converge when $d$ grows larger. The biggest improvements on these three models are $7.26\%$, $7.76\%$ and $5.64\%$ respectively. Their average is over $6.5\%$ which is a huge progress.

### 6.2 CHINESE POEM GENERATION

The results of our BLEU-based evaluation are summarized in Table 2. We compare our **Ham-based PPG** with several relevant baselines like **Statistical Machine Translation (SMT)** proposed by He, et al. (2012) and **RNN-based Poem Generator (RNNPG)** proposed by Zhang, et al. (2014). In the former model, a poem is generated iteratively by translating the previous line into the next line. In the latter model, all the lines are generated based on a context vector encoded from the previous lines.

We also set different attention depths in order to learn the relationship between overall performance and the attention depth $d$.

| Model | BLEU$_1$ | | BLEU$_2$ | | BLEU$_3$ | | BLEU | |
|---|---|---|---|---|---|---|---|---|
| | 5-Char | 7-Char | 5-Char | 7-Char | 5-Char | 7-Char | 5-Char | 7-Char |
| SMT | 0.056 | 0.124 | 0.052 | 0.150 | 0.054 | 0.176 | 0.054 | 0.150 |
| RNNPG | 0.058 | 0.187 | 0.062 | 0.210 | 0.067 | 0.207 | 0.062 | 0.202 |
| PPG | 0.061 | 0.185 | 0.069 | 0.193 | 0.073 | 0.198 | 0.068 | 0.192 |
| 5-level **Ham** PPG | **0.063** | 0.210 | 0.070 | 0.237 | 0.075 | 0.226 | 0.070 | 0.224 |
| 10-level **Ham** PPG | 0.062 | 0.217 | **0.075** | **0.267** | **0.076** | 0.259 | **0.072** | 0.244 |
| 20-level **Ham** PPG | 0.062 | **0.221** | 0.074 | 0.258 | **0.076** | **0.260** | 0.071 | **0.246** |

Table 2: BLEU-based evaluation results

When attention depth $d$ is not large enough, the larger $d$ is, the better performance our model will achieve. **PPG** with 10-level **Ham** has almost 25% of improvement on the averaged BLEU score compared with ordinary **PPG**. On the other hand, through the last two rows of the table above and Theorem 2, we know that with the continuing increase of $d$, our performance will converge sooner or later. So in order to balance between performance and training cost, we suggest attention depth $d$ to be between 5 and 10.

| 暖冬 | 图形学 | 深度学习 |
|---|---|---|
| **Ham** | **Ham** | **Ham** |
| 占得诗人独处时， | 天上浮云水上云， | 一室虚空事已空， |
| 一江春水绿涟漪。 | 扁舟相竞逐行行。 | 不知何处寄幽踪。 |
| 朝来玉殿笙歌舞， | 幽人独倚阑干处， | 胸中只有千千寿， |
| 且向西风夜雨归。 | 只是无愁一片声。 | 记得诗人字字中。 |

Figure 3: Three quatrains generated by **10-level Ham PPG** model

# 7 CONCLUDING REMARKS

In this paper we have developed **Hierarchical Attention Mechanism (Ham)**. So far as we know, This is the first attention mechanism which introduces hierarchical mechanisms into attention mechanisms and takes the weighted sum of different attention levels as the output, so it combines low-level features and high-level features of input sequences to output a more suitable intermediate result for decoders.

We tested the proposed model **Ham** on the task of Chinese poem generation and machine reading comprehension. The experiment revealed that the proposed **Ham** outperforms the conventional models significantly, achieving the state-of-the-art results. In the future, we would like to study more applications of **Ham** on other NLP tasks such as neural machine translation, abstractive summarization, paraphrase generalization and so on.

Recall that **Ham** belongs to soft attention where every token of input sequences is calculated by attention function. We will extend **Ham** to hard attention and local attention, to show whether the performance can be better and whether **Ham** can fit reinforcement learning environment better. Also, we will attempt to extend **Multi-head Attention Mechanism** of Vaswani, et al. (2017) to its hierarchical version and apply to neural machine translation.

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
