# OpenReview forum: "Hierarchical Attention: What Really Counts in Various NLP Tasks"
_ICLR.cc/2019/Conference_

### Official Review · AnonReviewer1 · 2018-11-03
**A simple extension of multi-level attention, but needs more extensive comparison to existing methods**

**Rating:** 4
**Confidence:** 3

**Review:**

The paper introduces hierarchical attention, where they propose to weighted combine all the intermediate layers of multi-level attention. The idea is simple and seems to be promising, however originality seems incremental.

In order to fully demonstrate the significance of the proposed algorithm, the authors should conduct more comparisons, for example, to multi-level attention. Just comparing with one-level attention seems unfair given the significant increase of computation. Another aspect of comparison may be to consider computation and performance improvements together and discuss the best trade-off. The authors should also include some standard benchmark datasets for comparisons. The current ones are good but it is not so clear what is the best state-of-the-arts results on them when compared with all other methods.

The analysis on the network's representation and convergence is nice but it does not bring much insights. The argument for decreasing global minimal of the loss function in terms of increasing parameter size can be made for nearly all models but it is of little practical use since there is no guarantee one can reach the global optimal of these models.

I recommend the authors to analyze/demonstrate how effective this weighted combination is. For example, the paper can benefit from some clear examples that show the learned weights across the layers and which ones are more important.

The presentation of the paper needs some polishing. For example, there are numerous typos, grammatical errors everywhere.

---

### Official Review · AnonReviewer2 · 2018-11-04
**A few major issues**

**Rating:** 3
**Confidence:** 5

**Review:**

The paper proposes to enhance existing multi-level attention (self-attention) mechanism by obtaining query and key vectors (= value vectors) from all levels after weighted-averaging them. The paper claims that this is also theoretically beneficial because the loss function will converge to zero as the number of layers increase. It claims that the proposed architecture outperforms existing attention-based models in English MRC test (SQuAD), Chinese MRC test, and Chinese poem generation task.

I find three major issues in the paper.

1.  I think the proposed hypothesis lacks the novelty that ICLR audience seeks for. Through many existing architectures (ResNet, ELMo), we already know that skip connection between CNN layers or weighted average of multiple LSTM layers could improve model significantly. Perhaps this could be an application paper that brings existing methods to a slightly different (attention) domain, but not only such paper is less suitable for ICLR, but also it would require strong experimental results. But as I will detail in the second point, I also have some worries about the experiments.

2. The experimental results have problems. For English MRC experiment (SQuAD), the reproduced match-LSTM score is ~10% below the reported number in its original paper. Furthermore, it is not clear whether the improvement comes from having multiple attention layers (which is not novel) or weighted-averaging the attention layers (the proposed method). BiDAF and match-LSTM have single attention layers, so it is not fair to compare them with multi-layer attention.

3. Lastly, I am not sure I understood the theoretical section correctly, but it is not much interesting that having multiple layers allow one to approach closer to zero loss. In fact, any sufficiently large model can obtain close-to-zero loss on the training data. This is not a sufficient condition for a good model. We cannot guarantee if the model has generalized well; it might have just overfit to the training data.

A few minor issues and typos  on the paper:
- First para second sentence: In -> in
- First para second sentence: sequence to sequence -> sequence-to-sequence
- Second last para of intro: sentence fragment
- Figure 3: would be good to have English translation.

---

### Official Review · AnonReviewer3 · 2018-11-05
**Lacks Novelty , incomplete results**

**Rating:** 4
**Confidence:** 4

**Review:**

Overall, this is an incremental paper.
The authors propose a hierarchical attention layer, which computes an aggregation of self attention layer outputs in the multi level attention model. This seems like a small improvement.

There are results using this hierarchical attention layer instead of the vanilla attention layers on Machine Reading Comprehension and Chinese Poem Generation. The authors should have also included results on more tasks to show the clear improvement of the proposed method.

The issues with this paper are:
- Aggregating weights of different layers has been an idea explored before (Elmo, Cove, etc.). So the model improvement itself seems small.
- Lack of strong experimental evidence. In my regard, the experiments are somewhat incomplete. In both the tasks, the authors compare only the vanilla model (BIDAF, MatchLSTM, R-NET) and the model with HAM layers. It is not clear where the improvement is coming from. It would have made sense to compare the number of parameters and also, using the same number of vanilla attention layers  which outputs the last layer and compare it to the one proposed by the authors.
- Since the argument is towards using weighted average rather than the last layer, there should have been a more detailed analysis on what was the weight distribution and on how important were representations from different layers.

---

### Public Comment · (anonymous) · 2018-10-01
**Simple yet intuitive modification to Attention mechanism gives impressive results**

The papers presents a fairly simple addition to the prevailing attention mechanism in form of an attention layer along the depth. The authors test the new architecture on a couple of important NLP tasks and beat the existing state of the art approaches. The paper is clearly written and easy to follow, though the formatting and grammar could be improved.

---

### Meta-Review · Area_Chair1 · 2018-12-14
**Incremental work, with limited experimental validation**

**Confidence:** 5
**Recommendation:** Reject

**Metareview:**

The authors propose a hierarchical attention layer which combines intermediate layers of multi-level attention. While this is a simple idea, and the authors show some improvements over the baselines, the authors raised a number of concerns about the validity of the chosen baselines, and the lack of more detailed evaluations on additional tasks and analysis of the results. Given the incremental nature of the work, and the significant concerns raised by the reviewers, the AC is recommending that this paper be rejected.